# Identification of distinct tumor cell populations and key genetic mechanisms through single cell sequencing in hepatoblastoma

Alexander Bondoc [1,7✉], Kathryn Glaser [1,7], Kang Jin [2,3,7], Charissa Lake[1], Stefano Cairo [4,5], James Geller[6], Gregory Tiao[1] & Bruce Aronow [2,3]

Hepatoblastoma (HB) is the most common primary liver malignancy of childhood, and molecular investigations are limited and effective treatment options for chemoresistant disease are lacking. There is a knowledge gap in the investigation of key driver cells of HB in tumor. Here we show single cell ribonucleic acid sequencing (scRNAseq) analysis of human tumor, background liver, and patient derived xenograft (PDX) to demonstrate gene expression patterns within tumor and to identify intratumor cell subtype heterogeneity to define differing roles in pathogenesis based on intracellular signaling in pediatric HB. We have identified a driver tumor cell cluster in HB by genetic expression which can be examined to define disease mechanism and treatments. Identification of both critical mechanistic pathways combined with unique cell populations provide the basis for discovery and investigation of novel treatment strategies in vitro and in vivo.

[1] Division of Pediatric General and Thoracic Surgery, Cincinnati Children's Hospital, Medical Center, Cincinnati, OH, USA. [2] Division of Biomedical Informatics, Developmental Biology, and Pediatrics, Cincinnati, Children's Hospital Medical Center, Cincinnati, OH, USA. [3] Department of Biomedical Informatics, University of Cincinnati, Cincinnati, OH, USA. [4] Research and Development Unit, XenTech, Genopole—Campus 3, Fontaine, France. [5] Istituto di Ricerca Pediatrica (IRP), Corso Stati Uniti, Padua, Italy. [6] Division of Oncology, Cincinnati Children's Hospital Medical Center, Cincinnati, OH, USA. [7] These authors contributed equally: Alexander Bondoc, Kathryn Glaser, Kang Jin. ✉email: alex.bondoc@cchmc.org

Hepatoblastoma (HB) is the most common primary hepatic malignancy of childhood. The incidence of this tumor is 1.5 cases/million population/year or approximately 1% of cancers for young children, but has been increasing over the last 30 years[1]. Current standard of care is multimodal and includes neoadjuvant or adjuvant chemotherapy in conjunction with surgical resection or liver transplantation[2–6]. To understand the pathogenesis of HB, investigators have characterized the molecular signatures of HB using whole genome, exome, and RNA sequencing platforms[7–11]. Many signaling pathways have been implicated in the development of HB, including WNT, HH, NOTCH, YAP, and PI3K, but precise mechanisms are still unclear[12–14]. PDX models are derived from unique human tumors[15–17], permitting nonhuman treatment trials designed using clinically approved agents. PDX models can also aid in the identification of novel treatments based on the specific molecular biology of patient tumors[18]. We present here, HB PDX models mirroring the human disease, including aberrant gene expression allowing examination of the key drivers of tumor growth and proliferation. The current study evaluates patient source tumor, background liver, and PDX tumor to define tumor cell populations and key molecular pathways of HB that drive cancer growth and proliferation.

From a mechanistic standpoint, the WNT pathway has been implicated in the formation of pediatric liver cancer including HB[7]. The most common genomic alteration occurs in beta-catenin, so numerous studies have evaluated the importance of the WNT pathway and family of genes as potential therapeutic targets[7,19].

Additionally, Cairo demonstrated that HB with a proliferative phenotype, classified as C2 subtype, has a less favorable prognosis and resistance to treatment[7]. Further classification of this high risk group further defines C2a and C2b subtypes based on vimentin expression, and shows that patients with C2b tumors have a better outcome[20]. The patient tumors used for this investigation are histologically heterogeneous and treatment resistant with features of C1, C2a and C2b subtypes, and our PDX models demonstrate exaggeration of the parent tumor proliferation rate. Also of note, chromatin remodeling has been implicated in many malignancies, specifically resistance to chemotherapy[21,22]. In agreement with prior reports our single-cell RNA sequencing results demonstrate an important role of chromatin remodeling in HB tumor and PDX.

Together, WNT and PI3K/AKT signaling, in conjunction with cell cycle progression, morphogenesis, and chromatin remodeling in distinct tumor cell populations, may define the molecular biology of HB initiation, vascularization, maintenance, and tumor progression. The current literature does not clearly define the key driver cells of HB, but others have begun to explore driver genes as well[23]. There are reports to suggest that cancer stem cells play a role, but it remains unclear[24,25]. Additional reports in HCC implicate hepatocytes and microenvironment in tumor transformation, but still more investigation is warranted[26]. Our investigation defines clusters of cells with genetically defined roles, including a unique tumor-driving cell population not previously defined. Tumor subpopulations and distinct genetic signatures may be key to unraveling the factors driving liver cancer.

## Results

**PDX model mimics HB tumor expression and histology.** In the PDX models we have generated, patient tumors share genetic and histologic characteristics with PDX tumors. Figure 1a and b demonstrates the experimental workflow from primary tumor and liver collection through pathway and tumor subclustering analyses. We observed upregulation of GPC3 gene expression, a known marker, in HB tumor relative to background liver and nontumor liver (Fig. 1c). RNA sequencing (RNAseq) was performed on two HB patient tumors, background liver and PDX tumor over multiple passages (F0–F5) to evaluate maintenance of HB tumor phenotype. Evaluation of raw gene transcripts per million for key upregulated genes with strong upregulation in tumor and PDX, and prior reported involvement in HB, including GPC3, DLK1, and IGF2, showed robust increase in HB tumor gene expression with further elevated expression in PDX tumors (Fig. 1c). Using histological staining, we show an elevation of hematoxylin staining in HB primary tumor as well as tissue disorganization, characteristic of HB. Both primary tumor and PDX of HB17 and HB18 show cells that resemble developing liver consistent with the epithelial (fetal/embryonal) histology observed clinically (Fig. 1d, Table 1). A similar staining pattern between HB primary tumor and PDX was observed, as well as highly proliferative cells as indicated by immunohistochemistry (IHC) staining for KI67 (Fig. 1d). PDX samples show elevation of hematoxylin staining over primary tumor as well as prior observed histologic features in tumor (Fig. 1d).

GPC3 upregulation was also seen at the protein level by western blot with some variation in protein expression and presence of heparin sulfate (HS) side chain or 40-kDa cleavage product for some HB tumor and PDX samples (Fig. 1e). Interestingly, for both HB17 and 18, patient tumor shows more pronounced HS side-chain band than PDX tumors. Tumor characteristics are often exaggerated in the PDX model and some with passage (Fig. 1c, e, f). H&E and GPC3 histology show more a robust cancer phenotype in PDX tumor models. GPC3 is also elevated in tumor tissue by IHC, which is a clinical hallmark of HB tumor and more pronounced in PDX (Fig. 1f).

Table 1 outlines the clinical features for each HB patient evaluated in this study. Of interest, the HB tumors that support PDX growth are more complex, including HB17, 18, 21, 23, and 30 (Table 1). HB53 is still under investigation to determine if PDX generation is possible. HB17, 30, and 53 were selected for further in-depth analysis based on poor treatment response and outcome or metastasis, as well as robust growth in the PDX model in two samples.

**RNAseq HB tumor phenotype is maintained, and gene expression exaggerated in PDX.** The heat map in Fig. 2a shows dramatic expression changes from background liver samples (HB17B, 18B) to patient tumor (HB17T, 18 T) to PDX (HB17, 18 F0–F5). In general, the expression pattern is conserved from tumor to PDX with some fundamental differences (Fig. 2a). To confirm RNAseq expression patterns, a panel of genes were evaluated in six background liver and HB tumors including HB18, which was included in the RNAseq evaluation. In general, upregulation of GPC3, YAP1, SHH, CTNNB1, AXIN2, and FANCD2 was observed with some variation from patient to patient, with greater variability seen in YAP1 and beta-catenin (Fig. 2b).

**Single-cell transcriptome landscape of HB.** Single-cell RNAseq was used to further characterize the HB and PDX tumor compared with background liver to determine integral pathways and cell signatures in HB utilizing HB17, HB30, and HB53 samples (the second passage (F2) was used for available PDX samples). After preprocessing and quality control (see "**Methods**"), we collected 67,111 cells across seven samples, including three tumors, two background livers, and two PDX tumors. The single nuclei RNA sequencing analysis workflow is graphically demonstrated in Fig. 1b. Clustering was done, and eleven distinct cell types were classified (Fig. 3a) with enriched pathways and

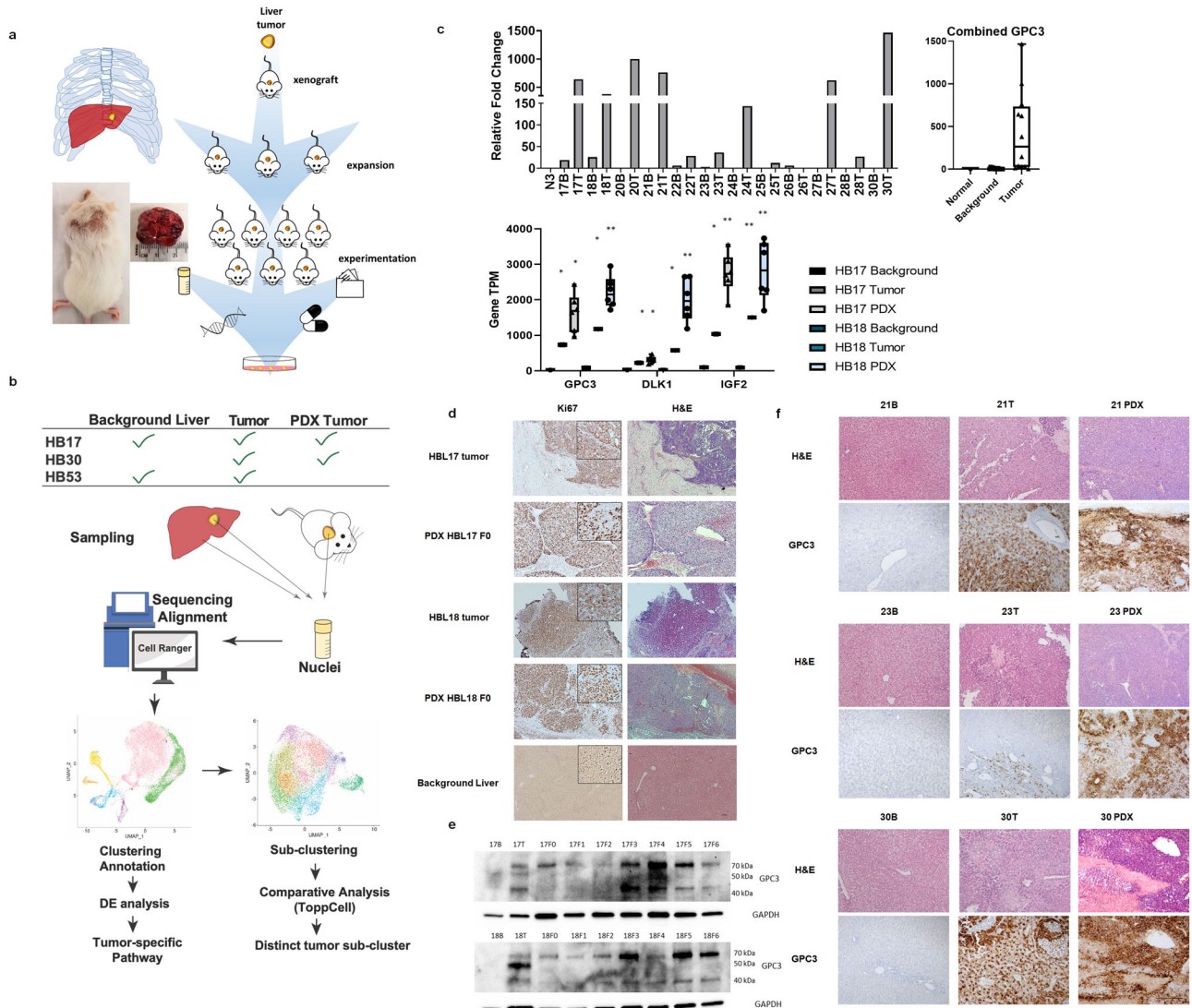

**Fig. 1 Comparison of HB background liver to tumor and PDX. a** Schematic of PDX model. Diagram showing the process of implantation of tumor, growth, and experimentation, including cryopreservation, molecular analysis, generation of cell lines, drug testing, and database of effective treatments. Mouse image demonstrates successful heterotopic tumor implantation with tumor growth. **b** Method and analysis workflow. Diagram of samples used for scRNAseq, and data analysis workflow through tumor sub-clustering (computer monitor image is a stock image from Biorender). **c** Real time PCR validation of *GPC3* gene expression of human HB background liver (B) and tumor (T) compared with control liver (N) (control liver = nontumor liver) (statistical analysis (student *t*-test) was performed on averaged data for N, B, and T samples, $p < 0.05$ for T vs N and T vs B. Each bar represents one patient sample). Raw gene transcript (TPM) values in background liver, HB tumor and PDX tumor showing increased expression of *GPC3* in tumor and PDX tumors (average of PDX's passage F0–F5) statistical significance determined by student *t*-test (SEM, *$p < 0.05$, **$p < 0.005$ relative to background of the same patient). TPM—transcripts per million. **d** PDX-HB17 and 18. Histology of primary tumor and PDX tumor (F0) of HB17 and HB18 and representative background liver (HB30) showing proliferation by H&E and KI67 (ma5-14520) (4x objective, scale bar 10 µM with zoomed-in 10 × 751 inlay, magnified 3x in KI67 panel). **e**. Western blot comparing GPC3 (ab207080) protein expression in B and T to PDX passages (passages = F0–F6). Tumor and PDX samples have elevated full-length, HS side, chain and cleaved N-terminal GPC3 fragments (the antibody does not detect C terminus). **f** Histology of background liver, primary tumor and PDX tumor of HB21, 23, and 30 showing maintenance of histologic features and proliferation by H&E, as well as increased GPC3 expression in tumor and even greater expression observed in PDX tumor (10x, scale bar 50 µM).

canonical markers, such as *CYP3A4*, *ALB*, *APOC3*, and *HPGD* for hepatocytes, *FLT1* for endothelial cells, *COL3A1* and *COL6A3* for hepatic stellate cells, *CD68* and *CD163* for Kupffer cells, and *PTPRC* for immune cells (Fig. 3b, c, Fig. S1). Hepatocytes are the dominant cell type in the background liver (Fig. S1), while tumor cells account for the largest proportion of cells in the HB tumor and PDX. We identified hepatic stellate cells, endothelial cells, NK cells, and T cells in both background liver and the HB tumor. Hundreds of mouse cells were found in the tumor environment of PDX and were then removed from analysis. The top 200

differentially expressed genes were identified and organized by sample, cell class, and gene module to show gene signatures that define the relevant cell types examined (Fig. 3c) with ToppCell (https://toppcell.cchmc.org/biosystems/go/index3/OncoMap) toolkit for each cell type (Methods). Further analysis, such as ToppGene enrichment analysis, as well as gene-interaction network, was also done in the integrated browser-able tool in the HB tumor. A heatmap of microenvironment cell gene expression demonstrates upregulation of cell activation, vascularity, and morphogenesis. Selected genes of cell-surface proteins, signaling

**Table 1 HB Patient Clinical Information. Staging, histology, diagnostic, and outcome data for HB patients evaluated in this study (follow-up is time from sample collection).**

| Patient # | Sample source | PRETEXT stage | Histology | Invasion on path | Metastasis | Patient outcomes | Follow-up (Days) | PDX model |
|---|---|---|---|---|---|---|---|---|
| 17 | Liver | N/A | HB—epithelial (fetal, embryonal), macrotrabecular, poorly differentiated small cell with anaplastic features | Negative margins, vascular invasion | None | Deceased | 139 | X |
| 18 | Liver | N/A | HB—epithelial (fetal & embryonal), mesenchymal without teratoid features, HCC-like; 50% viable | Negative margins, lymphovascular invasion | None | Alive without disease | 970 | X |
| 20 | Liver | IV | HCC—Grade 3-4 | Positive margin, vascular invasion, negative LN (0/1) | None | Deceased | 171 | |
| 21 | Liver | IV—P,E,F | HB—epithelial (embryonal), pleomorphic, HCC like; 80% viable | Focal positive margins, lymphovascular invasion, negative LN (0/3) | None | Alive without disease | 946 | X |
| 22 | Liver | III—F,C | HB—small foci of treated disease, no subtypes specified | Focal positive margin, LN negative (0/1) | None | Alive without disease | 773 | |
| 23 | Liver | III—P,V,C | HB—Mild pleomorphism, otherwise treated disease; 25% viable | Negative margins | None | Alive without disease | 835 | X |
| 24 | Liver | IV—V,P,E,F,C,M | HB—HCC-like, small cell undifferentiated, mesenchymal osteoid; 5–15% viable | Negative margins, focal tumor rupture, vascular invasion, LN negative (0/1) | Bilateral lungs | Alive with lung recurrence | 742 | |
| 25 | Liver | IV—V,P,F,M | HB—epithelial (fetal), osteoid; <10% viable | Focally positive margins with nonviable tumor, vascular invasion, LN negative (0/3) | Bilateral lungs | Alive without disease | 805 | |
| 26 | Liver | IV—F,C | HB—epithelial (embryonal, fetal), blastemal component; 15% viable | Negative margins, lymphovascular invasion, LN negative (0/1) | None | Alive without disease | 790 | |
| 27 | Liver | III—C,M | HB—epithelial (fetal, embryonal), osteoid; 45% viable | Negative margins, lymphovascular invasion, LN negative (0/1) | Bilateral lungs | Alive without disease | 742 | |
| 28 | Liver | III—M | HB—osteoid, epithelial (fetal), undifferentiated small cell; 15% viable | Margin positive, lymphovascular invasion, LN negative (0/2) | Bilateral lungs | Alive without disease | 762 | |
| 30 | Liver | IV—M | HB—epithelial (fetal, embryonal), transitional, small cell undifferentiated, HCC like, pleomorphic; 80% viable | Negative margins, lymphovascular invasion | Bilateral lungs | Alive with liver recurrence | 670 | |
| 31 | Liver | II—M | HB—epithelial (fetal, embryonal), mesenchymal, with blastema; 50% viable | Margin positive, lymphovascular invasion, LN negative (0/2) | Bilateral lungs | Alive with metastatic disease | 664 | |
| 38 | Liver | IV—P,V,F,C | HB—epithelial, mesenchymal with teratoid features. 50% viable | Negative margins, vascular invasion, LN negative (0/6) | None | Alive without disease | 557 | |
| 53 | Liver | IV—P,V | HB—epithelial (embryonal, pleomorphic) and HCC-like; focal mesenchymal with osteoid. 50% viable | Negative margins, vascular invasion, LN negative (0/5) | Bilateral lungs | Alive without disease | 269 | |

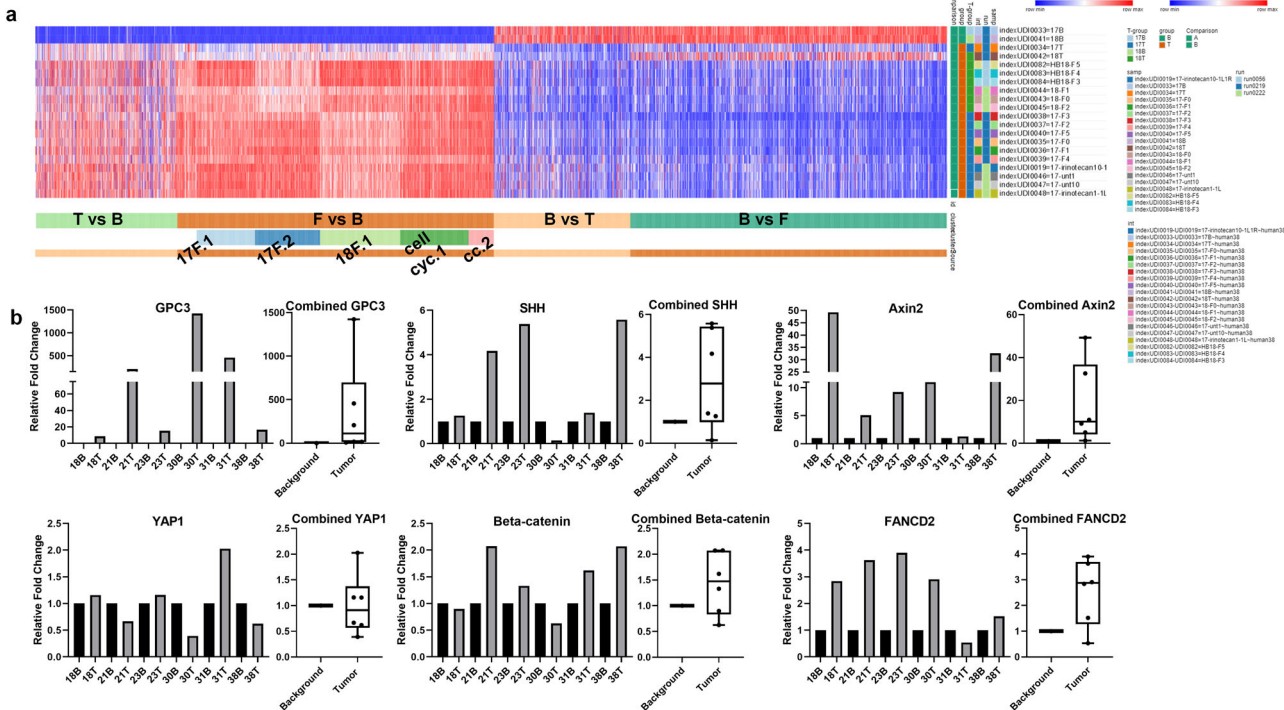

**Fig. 2 RNAseq comparison of HB samples. a** Heat map to demonstrate the expression differences between human background liver, source tumor, and PDX tumor over time (passages F0–F5 demonstrate expansion of tumor through multiple sets of mice). Overexpressed genes in HB tumor are often overexpressed to a greater degree in PDX and genes with lower expression can demonstrate further reduction in PDX. PDX tumor maintains a similar gene expression profile through multiple passages. **b** Validation of upregulation in genes of interest. qRT-PCR validation of gene expression in human HB background liver (B) compared with tumor (T). Each number—B/T represents matched samples from one patient. *GPC3* is overexpressed in T vs. B. *YAP1* shows some elevation in T, but expression is variable. *SHH* is generally overexpressed in T. Beta-catenin (*CTNNB1*) is variable but generally has moderate elevation in T. *AXIN2* and *FANCD2* are elevated in many T samples and appear to correlate with PDX tumor-growth success.

ligands, and transcription factors among differentially expressed genes (DEGs) in each cell type were used to build the interaction network in the tumor environment to show crosstalk with tumor cells. Tumor cells show interaction with the microenvironment, including endothelial cells through *DLK1* and *BMP* signaling, as well as tumor interactions with hepatic stellate cells as well as endothelial cells (Fig. S3a, c).

**PDX maintains features of the HB patient tumor at the single-cell level.** We compared gene correlation, cell cycle phases, upregulated genes, and enriched tumor-driven pathways across the HB tumor and PDX. The relationship between background liver, tumor, and PDX was evaluated by the correlation of gene expression. Tumor and PDX show high correlation ($R^2 = 0.902$) with background showing moderate correlation ($R^2 = 0.629$ and 0.540) (Fig. 4a, "Methods"). Not surprisingly, upregulated pathways in tumor and PDX highlight *WNT* signaling, cell cycle, organ development, and morphogenesis (Fig. 4a). The Volcano plots show similarities as well as differences between tumor and PDX. We investigated upregulated genes in the HB tumor and PDX and found that close to half of top DEGs were overlapping (Fig. S2b)[27]. PDX and HB tumor are different in some perspectives, for example, upregulated immune responses in tumor relative to PDX, which might be influenced by tumor microenvironment. This difference may be attributed to the lack of immune system in immunocompromised mice and the filtering of mouse cells removing the mouse microenvironment from this analysis (Fig. 4b). Upregulation of tumor signatures, such as *GPC3*, *DUSP9*, and *CTNNB1* can be observed in tumor cells and PDX (Fig. 4c). Predicted cell cycle scores (Fig. 4d)[28] show increased proliferation in both tumor and PDX samples. The

DEGs in the tumor cells of both HB tumor and PDX show significant enrichment for *WNT* signaling pathway and tissue morphogenesis (Fig. 4e). Besides those, tumor cells also exhibited high activities in *PI3K* signaling pathway and *C-MYC* signaling pathways, which have been proven to be closely related with HB development[7,29]. Functional associations were examined between genes upregulated in HB tumor and PDX, which demonstrated *WNT* signaling, cell cycle regulation, *PI3K* signaling, *C-MYC* signaling, and cell morphogenesis prominent in tumor cells (Fig. 4f).

**Single-cell gene expression defines distinct tumor cell clusters.** The single-cell RNAseq data were also used to examine cell clusters and explore the progression of gene expression state. Separate from the prior clustering of individual cell types, we reclustered the 52,629 tumor cells in HB tumor and PDX and identified 12 distinct clusters (Fig. 5a). Fig. 5b also shows the percentage of cells in each tumor cluster present in HB tumor and PDX. All clusters identified in tumor are present in some portion of the PDX tumor, except for Tr10, which is a cluster with a neuronal gene expression signature.

Utilizing the top differentially expressed genes critical upregulated pathways were defined to further characterize tumor cell clusters (Fig. 5c). Differentially expressed genes critical upregulated pathways were defined to further characterize tumor cell clusters (Fig. 5c). *GPC3* and *HMGA2* are upregulated in all tumor clusters compared with background liver. Other *WNT* pathway genes as well as anti-apoptosis genes, metabolic genes, and *PI3K/AKT* and *C-MYC* signaling pathways help to further suggest distinct roles of cell clusters within the tumor.

The distribution of known genes in HB, expression pattern of published signature genes, and the up- and downregulated

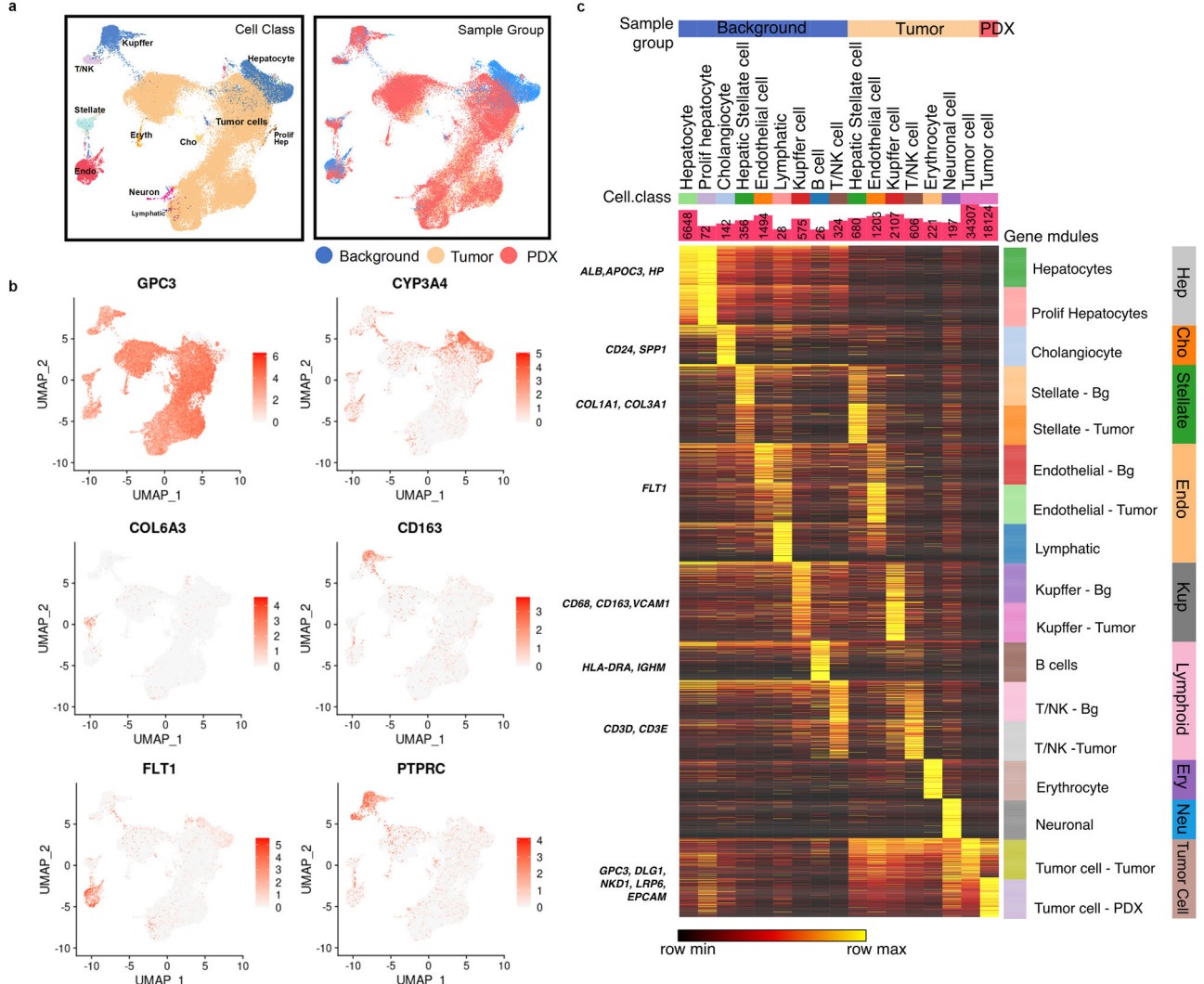

**Fig. 3 Single-cell RNA sequencing characterization of background liver, tumor, and PDX. a** UMAP of cell classes (Left) and sample groups (Right) of the integrated data from background, tumor and PDX. Endo endothelial cell, Stellate hepatic stellate cell, Eryth erythrocyte, T/NK 774 T cell or NK cell, Cho cholangiocyte, Prolif Hep proliferative hepatocyte. **b** Marker genes for common cell populations in liver and tumor were drawn on UMAP. *DLK1* shows tumor, *HPGD* identifies hepatocytes, *FLT1* indicates epithelial cells, *COL6A3* represents hepatic stellate cells, *CD163* marks Kupffer cells, and *PTPRC* indicates immune cells. **c** The heatmap for 200 most upregulated genes for cell types within background liver, tumor, and PDX (Methods). Known markers were shown on the left. Cell counts are shown in bar charts.

functions in key tumor cell clusters are shown in Figs. S3 and S4 to aid definition of cell clusters. The distribution of tumor types by cell cluster is shown in Fig. S3a to show that most clusters are present in tumor and PDX with cluster 10 being an exception.

Violin plots of the detected genes show the distribution of gene expression across the defined clusters and suggest that clusters 1 and 4 have lower UMIs, which may be due to lower quality (Fig. S3b). The gene expression distribution and location of known markers in HB, including *GPC3*, *VIM*, and *HMGA2*, are shown in Fig. S4a,b. The localization of cells from each patient tumor and PDX is visualized by UMAP to show the sample distribution and contribution of each patient tumor to our analysis. Each patient has a distinct tumor with numerous differences, but we observe an extensive overlap of the prominent clusters (Fig. S3c). To further understand the key defined tumor clusters (Tr0, Tr2, Tr3, and Tr5), the aberrant biological processes were defined from the top differentially expressed genes (Fig. S3d).

We examined protein expression in B, T, and PDX samples for CCND1, KI67, CD34, GPC3, YAP, and EZH2 to provide loose validation of different tumor clusters. CCND1 is expressed in all

tumor clusters with more seen in Tr1 and Tr4. KI67 is expressed in Tr2 and Tr8. CD34 has lower overall expression with the majority of expressing cells in Tr4 and Tr1. GPC3 is expressed in all tumor clusters and YAP1 is expressed in all clusters with a slight increase in Tr0. EZH2 is localized to defined clusters with expression most prominently in Tr2 (Fig. 5d, e). There are localized regions of overlapping KI67 and EZH2 staining, highlighting the Tr2 driver cell population with staining of GPC3 and YAP also in these regions of replication. The pattern of each protein localizes to different regions of the tumor and correlates with the gene expression level seen from scRNAseq, but additional validation is required (Fig. 5d, e).

Additional genes within key pathways, including *WNT*, *NOTCH*, *PI3K/AKT*, and *C-MYC*, are listed in Fig. S5, as well as cyclin-dependent protein kinase activity genes, highlighting the prominence of the overexpression seen in cluster Tr2. Tr2 has high expression of *FANCD2*, *RBL1*, *EZH2*, and numerous WNT pathway genes, Tr4/1 has elevated *FLT1*, *CALCRL*, *VIM*, and *VCAN*[30], Tr5 has high *PTCH1*, *LEF1*, and *CD44* expression, and Tr3 has elevated *EGFR*, *CYP3A5*, and *IGFBP2* among many other

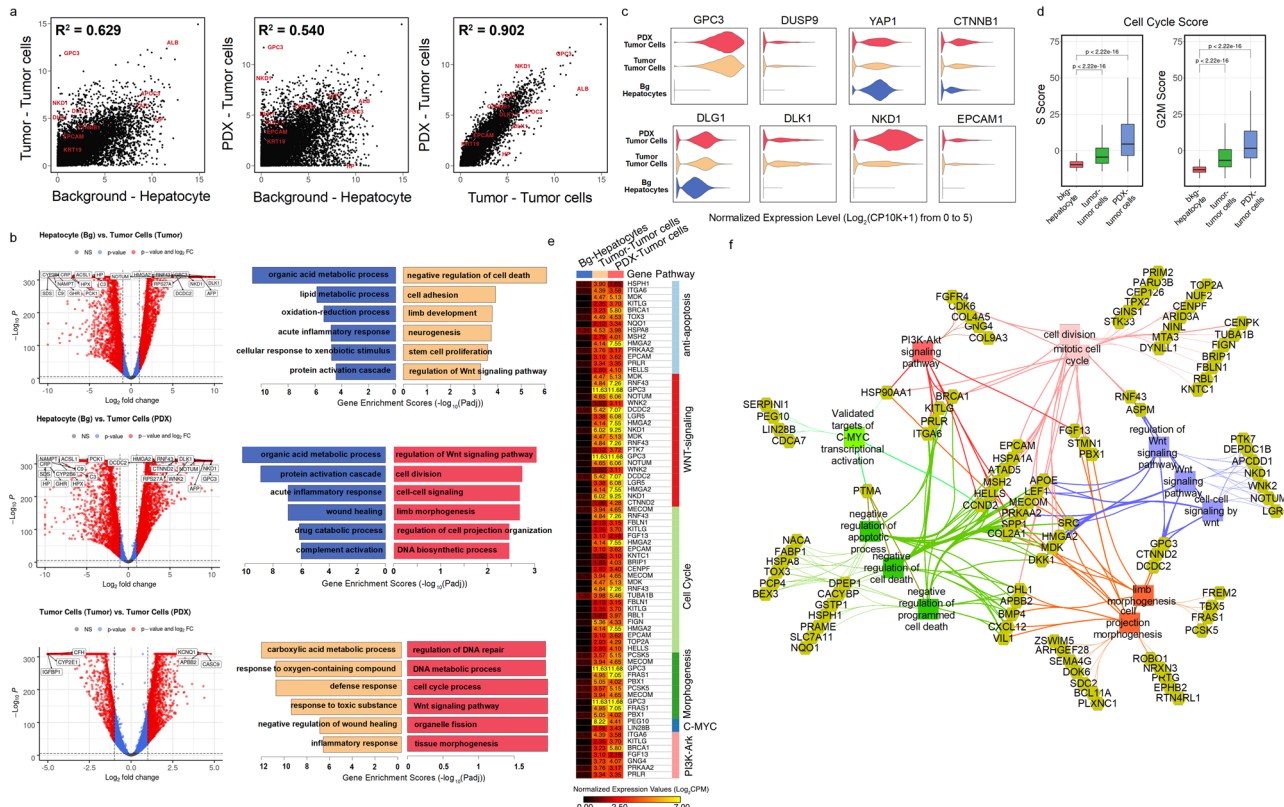

**Fig. 4 Maintenance of features across the tumor and PDX. a** Scatter plots for normalized expression values of all detected genes in hepatocytes of background liver versus tumor cells of tumor (Left), hepatocyte of background liver versus tumor cells of PDX (Middle), and tumor cells of tumor versus tumor cells of PDX (Right). R-square values were calculated after fitting data into linear-regression models. Some markers of tumor cell and hepatocytes are highlighted. **b** Volcano plots (Left) and gene enrichment results for top 200 DEGs (Right) of comparisons in (A). Representative enriched pathways were shown, and gene enrichment scores were calculated using −Log10 (adjusted enrichment p value). **c** Violin plots showed distributions of normalized expression levels of several important tumor genes across background hepatocytes and tumor cells in tumor and PDX. **d** Cell cycle scores, including scores of phase S (Left) and G2M (Right) of three main cell types showed that tumor cells had significantly higher proliferation activities. **e** Genes involved in the upregulated pathways of tumor cells in (B) were shown on the heatmap (Table S5). Normalized expression values were used. **f** Functional association network showing upregulated genes of tumor cells in HB T and PDX and their associations with tumor development-related pathways in (b and e) using ToppCluster (Methods).

defining expression patterns (Fig. 5c, Fig. S4). Tr0 cells have high expression of *WNT* and *NOTCH* genes but to a lesser degree than other clusters consistent with a more primitive or less active state. The similarity matrix of the 12 clusters identified in tumor shows how similar the clusters are and highlights prominent differences in the Tr0, Tr2, Tr4, Tr5, and Tr3 populations. Tr4 and Tr3 show the most notable transcriptome differences, but Tr4 also has lower read depth, which may indicate poor cell quality (Fig. 5c).

Utilizing gene expression data, a heatmap of upregulated functions was defined in tumor clusters from HB tumor and PDX. Distinct cell clusters display unique functional roles by pathway enrichment (Fig. 5f). Tr2 has high cell cycle function and replication, and also showed downregulation of other cell functions suggestive of an initiation state. Tr5 demonstrates a role in morphogenesis, as well as higher levels of cell migration-related genes. Tr3 has a high metabolic role, with Tr11 also having an increase. Tr4/1 has prominent expression in response to stimulus, immune regulation, apoptosis, and secretion as well as others as shown in Fig. 5f[31]. Tr0, Tr2, Tr6, Tr7, Tr8, and Tr9 share similarity to many other clusters. Tr11 has similarities to both Tr0 and Tr3 (Fig. 5c). Examination of pathways, transcription factors, and cluster gene expression data (Fig. 5c, e, f) support an initiating/driving role of Tr2 through gene expression, including *EZH2* and *RBL1*, and upregulation of cell cycle and cell division, tumor development for Tr5 through *LEF1*

and *PTCH1* expression and increased morphogenesis and development, loose support for endothelial nature of Tr4 through *DLK1*, *FLT1*, and *VIM* signaling, and tumor maintenance for Tr3 through strong metabolic signature and *EGFR* and *IGFBP2* expression. To aid in definition of tumor subtype and prognosis, a genetic signature was previously established. A heatmap was generated to show the tumor cluster expression of the 16 gene signatures used predominantly in the field of HB to define tumor subtype. Interestingly, the expression profile generally corresponds to previous reports with varied expression in some clusters (Tr3, Tr4, and Tr5) (Fig. S4d)[7]. Heatmaps of the individual sample gene expression signatures are shown in Fig. S6, to highlight both the consistent and variable expression inindividual tumor samples. Core proliferative signatures are well preserved across samples, while individual patient differences are still observed in replicating and nonreplicating tumor cells. The key features that are specific to each individual will provide the framework of future studies.

**Identification of cell cluster progression and tumor cell cluster targets.** Utilizing UMAPs of HB tumor and PDX, RNA velocity was predicted with scVelo to extrapolate differential flow from one cell cluster to the next[32,33]. RNA velocity shows little movement in central clusters, with flow down from Tr0, which

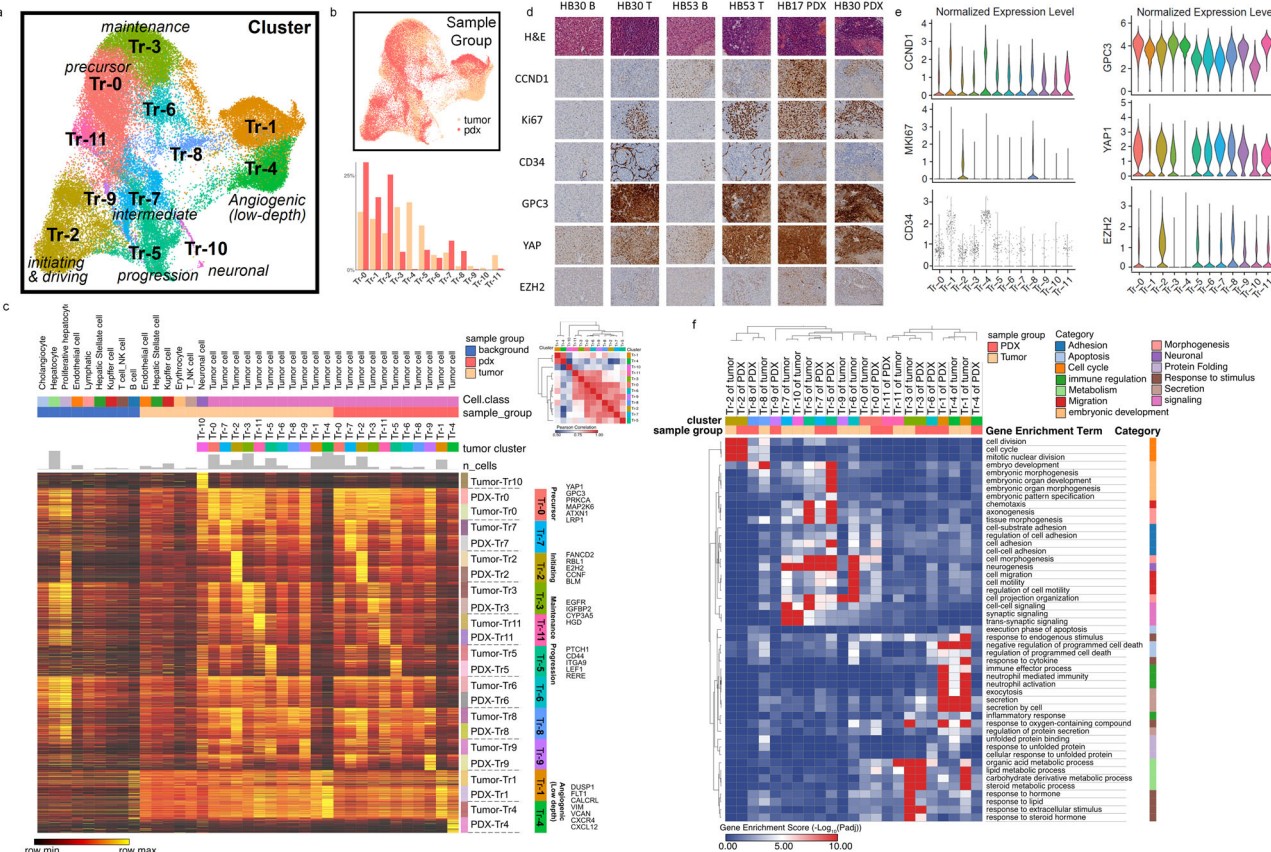

**Fig. 5 Gene expression defines initiating cell potential and other functional roles in tumor cell clusters. a** The UMAP visualization showed the clusters found in the tumor cells from tumor and PDX after integration and reclustering. Cells in different states are labeled. Tr1 and Tr4 are clusters with relatively low sequencing depth. **b** UMAP visualization of sample-group distribution (Top) and percentages of clusters in each sample group (Bottom). **c** A heatmap was drawn for 200 most upregulated genes per tumor cluster in each sample group. Cell types in background liver and tumor microenvironment are shown as the reference. Normalized values were used for expression levels. Key signature genes in some clusters are shown on the right. Similarity matrix of tumor clusters was calculated with Pearson correlation based on a combined gene list from (c). Hierarchical clustering was applied for rows and columns. **d** IHC was conducted in B, T, and PDX samples from HB30, B and T from HB53, and PDX from HB17 to show localization of CCND1, KI67, CD34, GPC3, YAP, and EZH2 to differing regions of tumor (20X, scale bar 50 μM). **e** Violin plots of gene expression for the corresponding genes in (d) are shown to show the expression patterns in each tumor cluster. Dots are only shown in the violin plot of *CD34* due to its relatively sparse expression in the single-cell data. **f** A heatmap of gene set enrichment scores ($-\log_{10}$Padj) of each cluster by ToppGene enrichment using genes in (c). Tumor-association enrichment terms in Gene Ontology (Biological Process) were selected and grouped into several categories. Hierarchical clustering was applied for rows and columns.

may be a precursor population to other clusters including Tr2 and 5 (Fig. 6a). Tr4 demonstrates flow into additional tumor cell clusters with Tr4 more prevalent in primary HB than in PDX. Tr1 has a similar signature to Tr4, but is derived from the male patient in this cohort. The significance of this variation is unclear. We propose based on gene expression, pseudotime trajectory, and RNA velocity that cluster Tr0 is a precursor cell, Tr2 is a potential initiating/driver population that differentiates in to Tr5 and Tr3 cell populations, and the Tr4 endothelial-type cluster works with other tumor cells to drive HB tumor (Fig. 6b). Tr3 also shows expression and velocity toward Tr0, which may suggest that this cluster has some place between normal liver and tumor initiation, but more data are needed.

UMAPs of tumor and PDX mapped with background liver further clarify the flow of tumor cells (Fig. 6a). Figure 6c displays a heatmap of potential driver and targetable genes in tumor cell clusters compared with cell clusters in background liver. Numerous genes have elevated expression in tumor clusters often with even greater expression in Tr2. This heatmap also highlights unique genes that are expressed only in more critical cell clusters. Network analysis was conducted to further evaluate driver genes prominent in Tr2 to identify treatment targets and

generate hypotheses for future studies (Fig. 6d). WNT signaling and cell cycle show prominent expression and interconnectedness within this network.

**Complex signaling network between tumor cells and the tumor microenvironment.** Further interactions were characterized by evaluating the network of signaling pathways between tumor cell clusters and tumor microenvironment cells (Fig. 7). The gene expression changes in each cluster and the differential signaling network demonstrate different functions of cell subclusters as they become more differentiated tumor cells that may support tumor growth and maintenance. The signaling patterns between subclusters and support cells were inferred using CellChat and highlight four patterns of expression. Pattern one and four highlight the expression changes from Tr0 to Tr5/7 supporting the RNA velocity seen in Fig. 6a (Fig. 7a). Prominent pathways present within these patterns are also defined and include *WNT*, *NOTCH*, *HH*, and numerous others. The interactions between each cluster are shown in Fig. 7b. Using SHG imaging, we further explored the vascular signature by visualizing the collagen in tumor compared with background, demonstrating a similar expression of collagen in both with vast disorganization seen in

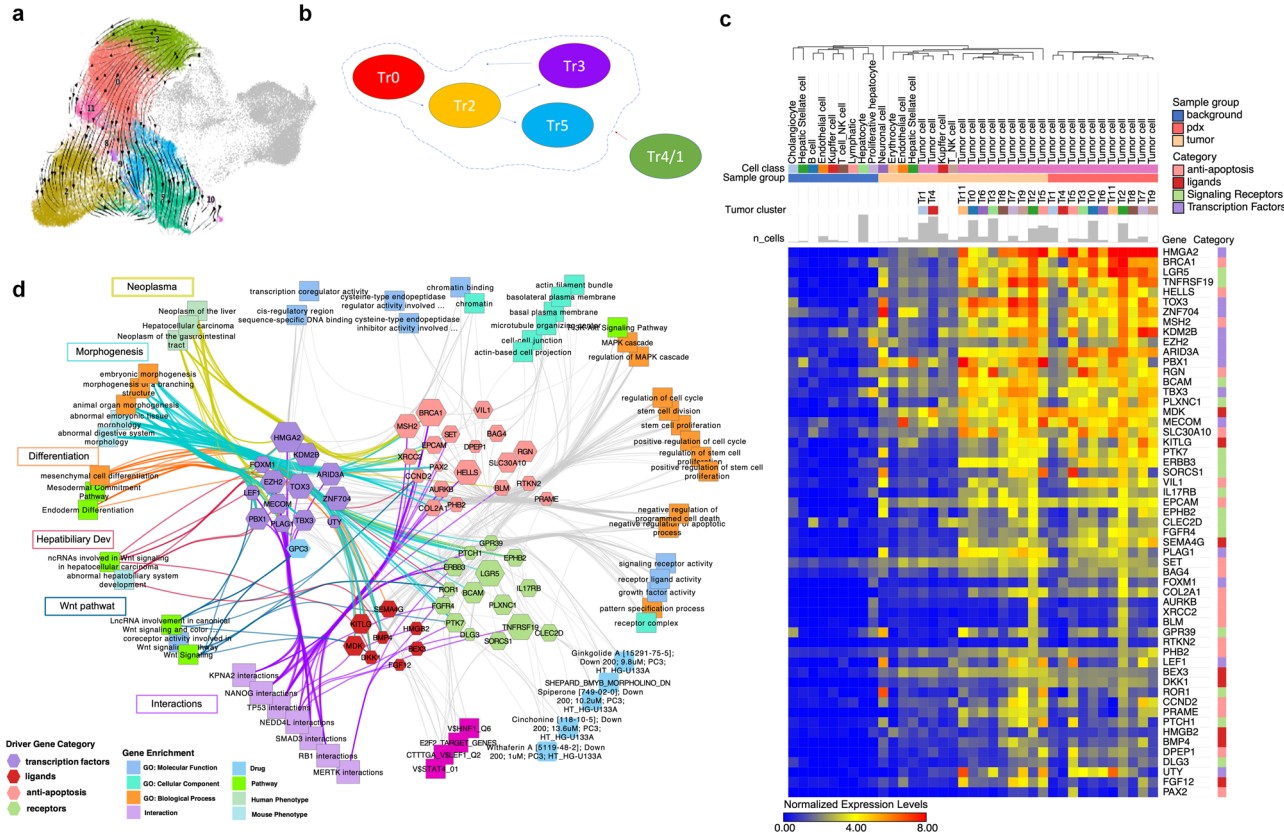

**Fig. 6 Definition of cell-cluster progression and molecular targets. a** RNA velocity of tumor clusters is shown on the UMAP. Streams indicate the predicted tumor transitions across clusters (Methods). We only focused on clearly identified clusters and removed clusters with low sequencing depth (Tr1 and Tr4) since RNA velocity is easily biased by technical noise[58]. **b** Diagram of proposed tumor cell cluster progression from precursor (Tr0) to initiating cells (Tr2) to progression (Tr5) and maintenance cells (Tr3) with integration of potential angiogenic/endothelial cell population (Tr4). **c** Heatmap of potential driver or targetable gene sets in tumor cells, background liver, and tumor environment. Differentially and highly expressed (maximal row levels are greater than 0.5) anti-apoptosis genes, ligands, and receptor genes and transcription factors in tumor cells are selected and shown (GO:0043066: negative regulation of apoptotic process; GO:0048018: receptor-ligand activity; GO:0004888: transmembrane signaling receptor activity; GO:0000976: transcription-regulatory region sequence-specific DNA binding). Endo endothelial cells, Cho cholangiocytes, HSC hepatic stellate cells, Kup Kupffer cells, B B cells, NK NK cells, T T cells, Hep hepatocytes, Mono monocytes, T/NK T cells or NK cells. **d** Network showing driver genes (**c**) and associations with tumor-related terms enriched in ToppCluster. The size of hexagon indicates the gene expression level in tumor cluster Tr2. Key functional associations, such as morphogenesis and Wnt signaling pathway, are highlighted in different colors.

the tumor samples consistent with immature vessels (Fig. 7c). Further exploration of complex pathways shows overexpression of receptors and ligands to promote signaling between tumor subclusters and the microenvironment (Fig. 7d). We have highlighted *BMP*, *FGF*, and *CXCL* signaling pathways due to the robust signaling network and the overexpression of ligands and receptors in clusters with potential to support tumor. BMP has classically been suggestive to be tumor repressive, but recent reports suggest a role in EMT[34].

Clusters Tr0, 6, 7, 10, and 11 have high expression of *BMP* ligands with vast expression of BMP receptors in most clusters and support cells. *FGF* signaling has been reported in numerous other cancers to promote growth and tumor progression[35,36]. In line with these reports *FGF2* is upregulated in Tr2, which has been reported to drive tumor and circulate in tumor microenvironment[36]. *CXCL* signaling is also of interest based on a reported role in metastasis and interaction with endothelial cells[37]. Our data show high expression of ligand in many tumor clusters, including high *CXCL12* in Tr11, *CXCL2* in Tr3, and *CXCL16* in Tr10 with high *CXCR4* receptor expression in tumor endothelial cells supporting prior reports (Fig. 7d). By focusing on endothelial cell interactions, we offer further support that tumor cells are interacting to support tumor growth and

vascularity. There is a high level of differential signaling between the different cell subtypes, including tumor endothelial cells and prominent clusters.

## Discussion
Utilizing scRNAseq, we have clearly defined cell populations from tumor and PDX from HB patients in contrast to the cell clusters representative of normal liver identified from background liver. Our data demonstrate preservation of prominent tumor features in PDX even though heterotopic implantation does not wholly preserve the tumor microenvironment. The supporting cells that were isolated from tumor show differences from cells in background liver, particularly endothelial cells and NK/T cells. These differences seem to indicate a greater degree of activation in the tumor microenvironment. Although PDX models, including our subcutaneous model, have limitations such as lack of tumor microenvironment, the preservation of key driving pathways demonstrates the value of this model system to recapitulate HB molecular signatures and serve as a model to conduct murine clinical trials to evaluate the effectiveness of current therapies, as well as discover and evaluate novel treatment strategies in real time. The gene expression data from HB PDX models also suggest

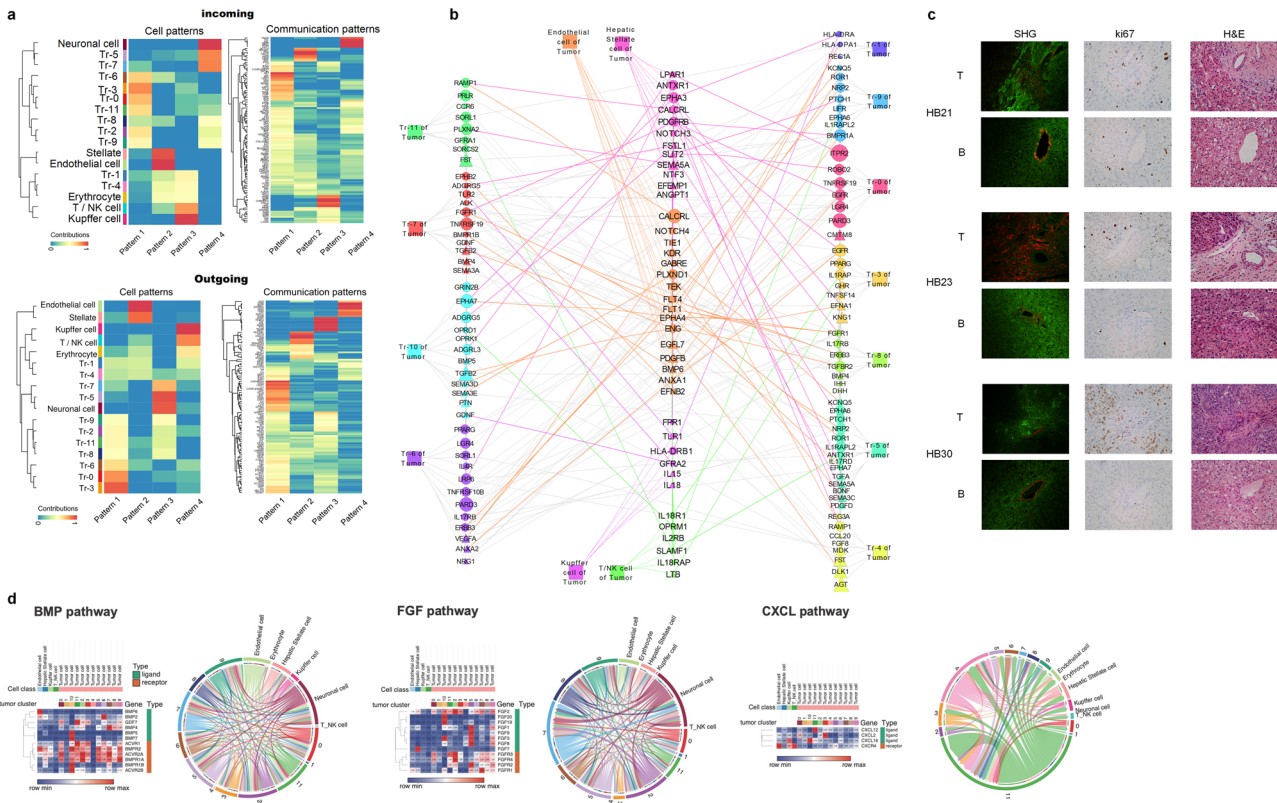

**Fig. 7 Differential signaling between tumor cell clusters and microenvironment cells. a** Signaling pathways were predicted using CellChat to show interaction patterns between tumor subclusters and microenvironment cells. Four main patterns of ingoing and outgoing signals were identified, and associated pathways shown. **b** A network of signaling interactions of upregulated genes between tumor cell clusters and supporting tumor microenvironment cells shows crosstalk between cells in tumor. Gene–gene interactions were inferred using ToppCluster and interaction pairs between tumor cells and microenvironment cells were highlighted. Node size represents normalized expression levels of genes in each cell type. Node shape represents ligands (triangle) and receptors (circle). **c** Second harmonic generation (SHG) microscopy of tumor and background liver. Left to right, SHG (red —SHG, green—autofluorescence), KI67, and H&E staining demonstrates disorganized vessel structure in tumor samples. 20X Plan APO, 1024 × 1024 resolution, scale bar 50 μM. **d** Heatmaps and CellChat chord diagrams of upregulated pathways with indicated receptor/ligand interactions are highlighted. Extensive *BMP*, *FGF*, and *CXCL* signaling networks are shown. Lowly expressed genes in each pathway (maximal row values less than 0.5) were removed from the heatmap.

that *AXIN2* and *FANCD2* overexpression signatures may serve as markers for successful PDX generation.

The tumor clusters have gene signatures, which indicate cell populations with five distinct phenotypes: precursor, initiating/ driving, endothelial-like, progressive, and maintenance cells. Our tumor and PDX samples demonstrate the presence of a potential initiating or driver cell population (Tr2), as well as distinct tumor cells, which were further subdivided into clusters with varying genetic signatures to highlight genes and pathways, which may serve as future treatment targets. The distinct cell populations that have been identified in tumor show known genetic signatures identified in HB tumor samples, but better suggest specific roles each tumor cell cluster may play in development and differentiation of HB. Our single-cell RNAseq data highlight clusters a global genetic signature consistent between source and PDX tumor, as well as initiating and transitional cell features including replicative capability that can support tumor growth and stability. Specifically, increased expression of *GPC3*, *DLK1*, and *HMGA2* are genes that define the driving pathways of HB, and these genes have also been previously reported as cancer stem cell markers in HB[7,38]. Together, these and other overexpressed genes highlight the importance of the *WNT*, *NOTCH*, *C-MYC*, *PI3K*, and *MAPK* pathways. Replicative cells from tumor are preserved in PDX demonstrating the presence of a potentially tumor-initiating population (Tr2)[39,40]. The gene signature of this cell cluster

includes *HMGA2*, *EZH2*, and *FANCD2*, as well as robust increase in cell cycle. These markers can be found in other tumor clusters as well but with less prominence. Tr2 cells may be the beginning of tumor formation, which gives rise to the signatures found in Tr5 and Tr3. There is also evidence of regional variation of replicating cells by KI67, which supports the potential of an initiating role for the Tr2-replicating cell cluster. Based on genetic profile and upregulated pathways, we suggest that Tr5 has a role in tumor progression and Tr3 has a role in tumor maintenance. The signaling of *CXCL12/CXCR4* may implicate Tr11/3 as having metastatic potential as well. Cluster Tr0 has features resembling a type of cancer stem cell or primitive driver cell with expression of most markers in other clusters but to a lesser degree with less pronounced cell cycling. Our data suggest maturation of distinct tumor clusters from initiating tumor cells into postmitotic support cells with roles in morphogenesis, adhesion, and vasculogenesis. Additional validation is required to determine if replicating cells drive tumor progression, and if nonreplicating cells can also differentiate into tumor driver cells. Lineage-tracing studies may clarify this question.

Additional evaluation in other patient tumors as well as in vitro studies is needed to clarify and validate these results, but there seems to be a transition from normal liver to Tr0/Tr2, with transformation to additional clusters being less clear, but Tr2 has a clear replicative/cell cycle role, and Tr5 is involved in migration

and morphogenesis. Tr3 has a metabolic role as well as greater response to the stimulus, and Tr0, Tr2, Tr7, and Tr9 shares roles with many other clusters. Tr4/1 has a unique signature and may be a key tumor endothelial population that helps tumor angiogenesis. It is also possible based on some differentiation and apoptosis markers that this population of cells is the result of stress response or cell death or simply a result of low read depth. These clusters suggest five distinct cell groups within the tumor with precursor, initiating/driving, endothelial, progressive, and maintenance phenotypes based on gene expression signatures[23,41]. The vast signaling network demonstrating communication between tumor subclusters and support for microenvironment cells demonstrates that tumor subclusters have extensive crosstalk to support the growth and differentiation of each unique cluster to maintain and protect the tumor. *FGF*, *BMP*, and *CXCL* signaling as well as numerous other pathways in Fig. 7 are key to formation of this stable interaction network within the tumor. These interactions will guide future exploration into the mechanism of how these clusters support vascularity and stemness within the tumor.

Our HB17, HB30, and HB53 tumors and PDX models have some genetic features consistent with all three (C1, C2A, and C2B) subtypes as previously described[20], highlighting the need to clearly define HB tumor cell functions. There are many gene expression differences between each patient tumor and between tumor and PDX, which require further exploration. These differences suggest a path for development of personalized medicine; however, our global analysis of multiple patient tumors highlights genetic similarities of potentially targetable cell clusters. This study is a step toward a more complete understanding of constituent HB cell subtypes and provides insight into a path forward to formulate more effective treatment algorithms based on cell signatures, as well as guide generation of additional model systems[42].

Targeting the upregulated genes in Tr2 and Tr5 as well as Tr3/11 may elicit a favorable response in tumors of this subtype, by preventing replication and reducing tumor structure and stability. Isolation and culturing of Tr2 cells and others based on the unique genetic signatures will allow examination of cluster tumorigenicity and inhibition. Pathways exaggerated in different clusters can be exploited to identify novel treatment candidates or combination therapies. As demonstrated in Fig. 6d, characterization of individual tumor complexity by scRNAseq may eventually and optimally provide real-time data to assist physicians in clinical decision-making but only after extensive investigation and clinical trials.

Examination of cell cycle in single cells demonstrates enhancement of proliferation in tumor and further exaggeration of proliferation in PDX. Enhanced proliferation and the progression of this increase in PDX models will allow these models of HB to more effectively guide identification of novel therapies for patients with relapsed or refractory disease. Given the unfavorable prognosis and chemoresistant phenotype demonstrated by HB tumors with the C2 and C2b subtype, which also defines the majority of samples from which we have successfully generated PDX models, having a model with even greater propensity to proliferate will allow for robust evaluation of the effectiveness of treatment options. These features of PDX will lend to characterization and analysis of tumor cell types, as well as better treatment outcome for patients in the future[43,44]. The identification of a tumor driver cell cluster (Tr2) as well as tumor-sustaining cell populations, predominantly Tr0, Tr3, Tr5, and Tr4/1, in primary HB tumor and PDX, will drive future studies to better understand how to control tumor growth and actively target the cancer-causing cells identified in HB tumor.

## Methods

**Human subjects**. We have collected background liver and tumor samples from patients with HB under conditions appropriate for characterization and further generation of PDXs with institutional review board approval (IRB # 2016-9497) and informed patient consent. Data from 15 patient tumors and PDXs generated from five of these HB patient samples are reported. Primary tumor and background liver were used to examine gene expression by real-time PCR, protein expressionby western blot and histology, and a small subset of these samples were used for bulk RNAseq and scRNAseq, as well as the corresponding PDX tumor samples. PDX tumor samples were also examined by western blot and histology. The workflow of this methodology is depicted in Fig. 1a, b. Primary tumor and background liver were used to examine gene expression by real-time PCR, protein expression by western blot and histology, and a small subset of these samples were used for bulk RNAseq and scRNAseq, as well as the corresponding PDX tumor samples. PDX tumor samples were also examined by western blot and histology.

**Animal studies, PDX generation, and monitoring**. We used female NOD SCID GAMMA C-/-(NOD.Cg-Prkdc*scid*Il2rg*tm1Wjl*/SzJ) (NSG) mice, 6–8 weeks old and surgically implanted human HB tumor heterotopically into the subscapular fat pad of the mouse[15,45,46]. Tumor growth was monitored, and volume was measured by caliper as described by Kats et al.[47]. All animal studies are completed with Institutional animal care and use committee (IACUC) approval and the following criteria defined by the "Guide for the Care and Use for Laboratory Animals" (IACUC # 2019-0077). Mice were fed irradiated diet and housed in a modular air-caging system with corncob bedding and nesting enrichment.

**Immunohistochemisty and second harmonic generation microscopy**. Immunohistochemical (IHC) staining of paraffin sections for hematoxylin and eosin (H&E), GPC3 (790-4564, 1:200 dilution; Roche), KI67 (MA5-14520, 1:300; Thermo Fisher), CD34 (AB8536, 1:200; Abcam), Cyclin D1 (790-4508, 1:200; Roche), YAP (AB52771, 1:200; Abcam), and EZH2 (3147, 1:200; Cell signaling) was performed on deparaffinized and rehydrated tissue sections with antibody diluted in phosphate-buffered saline (PBS) containing 2% BSA and 0.05% tween 20. Tissue sections were incubated overnight at 4 °C, washed with PBS + 0.05% tween 20, and then visualized with Vectastain elite ABC HRP (vector labs) per protocol instructions. IHC staining was performed in the Cincinnati Children's Hospital Research Pathology Core. Serial sections were imaged where indicated in the figure legends.

Second-harmonic generation (SHG) microscopy was conducting utilizing two photos to visualize fibrillar collagens[48,49]. SHG samples were deparaffinized and rehydrated and coverslips applied with PBS. Imaging was conducted on a Nikon FN1 confocal upright microscope with pulsed, IR laser at 1024 × 1024 resolution with 20x APO plan objective and 840-nm wavelength. Serial sections of tumor and background liver were used to capture SHG, KI67, and H&E staining.

**Real-time PCR expression and Western blot analysis**. Transcriptional gene expression was measured by quantitative reverse-transcription polymerase chain reaction. RNA was isolated from tumor and background liver tissue using the RNeasy Plus Mini Kit from Qiagen. cDNA was prepared using the VILO cDNA synthesis kit from Invitrogen. Polymerase chain reaction was performed on diluted DNA (1:20) using RT2 SYBR green master mix (Qiagen) in a CFX Connect real-time thermocycler (BioRad). RT2 primer assay gene primers were purchased from Qiagen and are listed in Table S1. Assays were performed in triplicate on each independent sample. For protein analysis, cell lysates were prepared using RIPA lysis buffer (Invitrogen) and aliquots were fractionated on gradient 4–15% polyacrylamide gels. Protein was blotted onto nitrocellulose and detected by the treatment of membranes with rabbit anti-human GPC3 (ab207080; Abcam) or mouse anti-human GAPDH (10R-G109A; Fitzgerald) and HRP-conjugated anti-rabbit (1706515; BioRad) or anti-mouse (1706516, BioRad) secondary antibody followed by ECL clarity HRP peroxide solution (BioRad). Viewing was accomplished with a ChemiDoc MP Imager (BioRad). All antibodies used are listed in Table S2.

**Bulk RNA sequencing**. RNA was isolated from tissue samples using Qiagen RNeasy Plus Mini kit following the manufacturer's protocol. HB17 and HB18 tumor, background, and PDX tumor from multiple passages were used for this analysis. About 150–300 ng of total RNA as determined by Qubit (Invitrogen) measurement was poly-A selected and reverse transcribed. RNAseq libraries were prepared using TruSeq polyA-stranded library preps from Illumina, and sequenced with paired-ends, with 100-bp parameters on the NovaSeq 6000 instrument. Sequences were aligned against GRCh38 and Ensembl annotation. Quality-control evaluation of the fastq files was performed using FastQC. RNA and nuclei in suspension were provided to the CCHMC DNA Sequencing and Genotyping Core for this analysis.

**Single-cell sequencing and read processing**. Background liver, tumor, and PDX tissues (25–30 mg each) were pulverized with liquid nitrogen and nuclei prepared, sorted, and counted as described by Wu et al. with addition of 0.04% BSA/PBS in the final buffer[50]. Three tumor samples (HB17, HB30, and HB53), two background

liver samples (HB17 and HB53), and two PDX tumors (HB17 and HB30) were used for this analysis. Single-cell RNA sequencing of nuclei was done by 10X Genomics with chemistry v3. FASTQ files were aligned to GRh38 reference genome and processed with Cell Ranger version 3.1.0 to obtain the unique molecular identifier (UMI).

**Preprocessing, integration, and clustering of single-cell data**. Cell ranger output of single-cell data was loaded using Seurat version 4 (3.9.9.9010). Cells with fewer than 500 expressed genes or 800 UMIs, or greater than 10% mitochondrial counts were removed. Single-cell data of background liver sample from HB30 were removed because of low quality. In the end, 67,111 high-quality cells were harvested from seven samples for downstream analysis. The total UMI counts per cell were normalized to 10,000 and the data were $\log_2$-transformed. Top 2000 highly variable genes were selected using the "vst" method of FindVariableFeatures function in Seurat. Data were scaled and principal component analysis (PCA) was conducted using highly variable genes with functions ScaleData and RunPCA, respectively. Principal components were used for integration of data from different samples using RunHarmony function of package Harmony[51]. Neighbors were found with top 30 components of Harmony-corrected cell embeddings using shared nearest-neighbor (SNN) graph implemented in the function FindNeighbors. Clustering was conducted using Louvain algorithm. Different resolutions, including 0.5, 1, and 2, were used to identify both general and fine-grained clusters. Uniform Manifold Approximation and Projection (UMAP) was calculated in the PCA space for visualization in the reduced dimensions. Cell classes were annotated for each cluster based on gene expression levels of known markers and markers from a human liver cell atlas[52] (Supplementary Data 1–3). Enriched pathways of differential expressed genes were also used for further cell annotations. Scanpy was used for stacked violin-plot visualization of marker genes[53].

**Tumor cell clustering**. Finer resolution of tumor subpopulations in the integrated data was achieved by extracting 52,629 annotated tumor cells from 5 HB tumor and PDX samples (HB17 tumor, HB17 PDX, HB30 tumor, HB30 PDX, and HB53 tumor). In order to remove batch effect between samples, we performed the reciprocal PCA procedure in Seurat, which is faster for large-dataset integration than the standard procedure. First, data were normalized, and highly variable genes were identified for each sample. Then variable features for the integration were selected using function SelectIntegrationFeatures. Then data were scaled, and principal component analysis was performed for each sample using integration-variable features. Integration anchors were found, and data were integrated using FindIntegrationAnchors and IntegrateData respectively. Then data scaling and principal component analysis were performed again on the integrated data. Top 30 principal components were used to find neighbors. Louvain clustering was done with FindCluster function under Resolution = 0.3. Among 11 identified clusters, clusters Tr1 and Tr4 have abnormally low UMIs and detected genes (Fig. S3), which might be low-quality cells. Upregulated genes and enriched pathways were used to define tumor clusters (Fig. 5) (Supplementary Data 4–7).

**RNA velocity**. Velocyto (PMID: 30089906) was used to measure spliced and unspliced transcripts in the sequencing data, which was further used for RNA-velocity computation in scVelo package[33]. Data normalization was done using pp.filter_and_normalize. Then, first- and second-order moments among nearest neighbors (n_neighbors = 30) in PCA space (top 30 principal components) using pp.moments. Stochastic model of RNA velocities was built using tl.velocity. Then velocities were projected onto a lower-dimensional embedding using tl.velocity_graph. Velocity flows were drawn using pl.velocity_embedding_grid.

**Cell cycle scoring**. Cell cycle scores, including S score and G2M score, were computed using the sum of scaled expression values of genes participating in S phase and G2M phase[54] (Supplementary Data 6). Student $t$-tests were used to evaluate the significance of cell cycle score difference between cell types.

**Transcriptional profile correlation evaluation**. Linear regression (lm function in R) was applied for mean-normalized expression values between cell types, including background hepatocytes and tumor cells in tumor and PDX (Fig. 4). Coefficient of determination ($R^2$) was computed after linear regression to evaluate transcriptional correlation between cell types.

**Differential gene expression and gene enrichment analysis**. Gene expression signatures for each cell type and subtype within each sample group (Fig. 3, Fig. 5) were computed from the Seurat-processed dataset using iteratively applied student $t$-tests embedded in the ToppCell portal (https://toppcell.cchmc.org/biosystems/go/index3/OncoMap). Besides, the same protocol was applied for tumor subclusters in each tumor sample for investigation of the individual tumor sample variation (Fig. S6). ToppCell provides easy user access to modularized representations of cell-type-specific gene expression signatures per sample group and provides a means of visualizing and carrying out post hoc analyses of the landscape of these modularized gene set signatures across all cell classes, subclasses, and samples and

sample groups. Default ToppCell signatures correspond to the top 200 genes of each comparison per sample group, cell class, and subclass. Gene modules are arranged hierarchically according to the cell annotations, each of which can be combined and compared for downstream biological annotation feature analysis using ToppGene and ToppCluster. More details can be seen in the tutorial of ToppCell. Additionally, the Scanpy function rank_genes_groups is used to compute differentially expressed genes between specific cell groups using student $t$ tests (Fig. 4b)[53]. Gene enrichment analysis was done by ToppGene to identify top enriched biological pathways, functions, and coregulatory associations. ToppCluster[55] was applied to compare different enrichment results of multiple gene modules across cell classes and tumor clusters. Various knowledge sources were used for enrichment, including gene ontology, as well as databases for interaction, drug, pathway, and human and mouse phenotype information. Adjusted $p$ values of enrichment were calculated using the Benjamini–Hochberg procedure. They were used for the computation of gene enrichment scores ($-\log_{10}(P_{adj})$), which evaluate the strength of associations between a gene list and an enrichment term. Volcano-plot visualization was generated using package *EnhancedVolcano*[56].

**Network analysis**. ToppCluster allows researchers to draw both functional enrichment networks and interaction networks. In our analysis, genes from different pathways (Fig. 4e) or various categories (Fig. 6d) were sent to ToppCluster for functional enrichment. The associations between genes and pathways in different groups formed into a network, which shows shared and exclusive genes in each group. Apart from the gene-pathway association network, we also inferred gene-interaction network in ToppCluster using ToppCell gene modules (Fig. 7, Fig. S2). Ligands and receptor genes were selected from gene modules and interactions were inferred based on curated-interaction database in ToppCluster. Further network analysis was supported by Cytoscape[57].

**Statistical evaluation and reproducibility**. Graphing and statistical calculations were performed using GraphPad Prism 8 software. Error bars denote standard error of the mean (SEM) and statistical significance was determined by unpaired, two-sided student $t$-test. Additional details are located in the figure legends. Data were confirmed in multiple human tumors, background liver, and PDX models to ensure robust and reproducible analyses.

**Biological material availability**. Patient-derived xenograft tumor is available if sufficient quantities are available based on successful tumor growth in vivo.

**Reporting summary**. Further information on research design is available in the Nature Research Reporting Summary linked to this article.

## Data availability
Sequencing data that support the findings in this study have been assigned Gene Expression Omnibus accession number GSE180666. Figures 1–7 and Figs. S1–S6 contain bulk or single-cell RNA sequencing data with associated GEO files. Single-cell RNA sequencing data analysis is in Supplementary Data 1–7. Raw data for real-time PCR and western blots (blots and analysis) are available in Supplementary Data 8–10.

## Code availability
The code used to analyze the datasets and draw figures is available at github: https://github.com/KANG-BIOINFO/scRNA-seq_Hepatoblastoma.

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

## Acknowledgements

We would like to thank the CCHMC DNA sequencing and genotyping core, the Potter lab, the Confocal Imaging Core, and the Research pathology core for technical expertize and support. This work was supported by NIH grant P30 DK078392 (Research pathology, DNA sequencing, Confocal, and genotyping core) of the Digestive Diseases Research Core Center in Cincinnati. Funding sources—AASLD Pilot award (SPR200430), Markham award, and Digestive Health Center: Bench-to-Bedside Research in Pediatric Digestive Disease (P30DK078392).

## Author contributions

A.B. contributed to the conception and design of the work, writing, analysis, interpretation, and critical review. K.G. contributed to the writing, conception, design, analysis, data interpretation and critical review. K.J. wrote portions of the methods and results sections, analyzed, and interpreted data and provided critical review of the paper. (A.B., K.G. and K.J. contributed equally as designated by[7]). Data collection was conducted by K.G., K.J. and C.L. S.C., J.G. and G.T. provided expertize in cancer, tumor modeling and critical review. B.A. was involved in data analysis, interpretation, and critical review of the paper. All authors provided final approval of the paper.

## Competing interests

The authors declare no competing interests.

**Additional information**

