## [Peer Review File · Communications Biology]

Reviewers' comments:

Reviewer #1 (Remarks to the Author):

The authors performed RNA sequencing and single-cell RNA sequencing to analyze gene expression patterns in hepatoblastoma tumor, patient derived xenograft (PDX), and background liver. The topic is of course interesting to the relevant community, and the data extremely valuable for future reference. However, I feel the data analysis and interpretation part needs to be improved before publication.

Scientific points:

(1) It will be helpful if the authors can also perform clustering analysis of all the cells, or just the tumor cells, from the PDX and the real tumor separately. This way we will see if PDX and real tumor indeed have similar features. More importantly, we can potentially see the subtle difference between the two as well.

(2) In Figure 4, it is good to see the tumor cells and PDX cells showed high correlation. Yet I think it is still worthwhile to look at the genes that are not correlated, and discuss potential reasons for the differences, that will be helpful for anyone using PDX model when they interpret their data.

Technical points:

(1) It will be good to provide more technical details, especially for the part covering cutting-edge scRNA sequencing data analysis. For example, the author mentioned certain cells are removed based on number of UMIs or MT reads, but without a discussion or reasoning what may occur to these cells that justify their removal. For another example, the authors said that the total UMI counts per cell were normalized to one million, without providing a raw UMI counts distribution in the cell population for the readers to visualize the data quality. Also by 10X Genomic platform it is possible to rely on RPKM as well, and there must be some reason to discuss why the authors chose to use UMI counts instead.

(2) When doing clustering, the authors claimed that 1 cluster has been removed among the 10 identified clusters, due to some abnormality. I wonder if it is possible to remove such abnormal cells first, and perform clustering later, to see if indeed the remaining 9 clusters will still exist. Sometimes the existence of abnormal cells during clustering analysis may affect the outcome of other clusters.

(3) Please provide more detailed info about how the authors performed clustering analysis, for the readers to make a well-informed judgement. Is it PCA or t-SNE? Is the outcome robust when the parameters or the sample size fluctuate?

Reviewer #2 (Remarks to the Author):

In this manuscript, Bondoc et al. provided an in-depth analysis of cell populations of hepatoblastoma (HB) by analyzing (single-cell) gene expression data generated from human tumor, background liver and PDX. Results showed that PDX was successfully established and several distinct cell populations were explored. Furthermore, the functional roles of these populations were discussed. Overall, this paper is easy to follow. I have two major comments which might strengthen the paper:

(1) The analysis of clonal analysis based on low-resolution CNV analysis (based on tools such as inferCNV) is lacking and the results might be very helpful in tracking the subpopulations from a perspective that well complements the GE-based clusters.

(2) Although the functional roles of identified tumor clusters were discussed, its translational implications are not studied. There are many bioinformatics tools can be used to deconvolute bulk tumor based on the gene expression signatures derived from scRNAseq. For example, do patients with more T5/6 signatures in the bulk tumor has worse outcome?

Reviewer #3 (Remarks to the Author):

Bondoc et al presented a single-cell analysis of a single case of Hepatoblastoma (HB) together with a small collection of patient tumors and PDX models. The authors identified cell populations within HB with specifically enriched pathways, suggesting possible treatment schemes targeting these populations. The idea and the approach are quite good and the analysis was well structured. However, this work has several important limitations which makes it hard to gauge the novelty of the study.

Major comments:

1) First of all, the number of cases is small (n=1) which makes it very hard to judge whether the pattern presented is unique to this special case or it is rather general. It will be important to increase the number of cases.

2) The authors presented a lot of results, many of which should have been explored by previous studies (e.g. bulk RNA analysis). There is not enough of link with previous work from the field and the novelty of the work is not high-lightened enough.

Detail comments:

3) The introduction of the manuscript is rather loose. The gap in the field and the scientific question is not very focused.

4) The experimental setup, especially the patient cohort and how they were used in different genomic survey (i.e. single cell or other approaches) is not easy to follow.

5) The authors tended to cherry pick different genes and pathways to interpret their findings. (e.g. GPC3, DLK1 and IGF2 on page 9, GPC3, YAP1 etc pathways on page 11). The logic behind the arguments is not so clear and it often made the presentation very punctuated.

6) The similarity between PDX and primary tumor is somewhat a controversial topic. There always will be some changes and mixed conclusions have been drawn for the differences. It will be good to draw the conclusion properly.

7) The most novel observation is Figure 6. However, the interpretation is very abstract. The pathways displayed in Figure 6D and Figure 7 are not explained clearly. How the evolutionary trajectory depicted in Figure 6B was derived is not explained properly.

With a bigger sample size, I think the study has the potential to draw a nice landscape of this important tumor.

We appreciate the review and recommendations of the reviewers and have prepared a comprehensive response to each point raised below. In summary, we have increased the sample size for our single cell RNA sequencing analysis and subsequently reanalyzed all data downstream which has allowed for a more robust analysis, but in doing so has required extensive revision to the manuscript and figures. Also, with addition of more background liver, tumor, and PDX samples the clustering was more complex, and we elected to rename the tumor subclusters to more clearly describe our results. Thank you for the constructive feedback.

Reviewer comment	Response to review
R1 Scientific points: (1) It will be helpful if the authors can also perform clustering analysis of all the cells, or just the tumor cells, from the PDX and the real tumor separately. This way we will see if PDX and real tumor indeed have similar features. More importantly, we can potentially see the subtle difference between the two as well.	Thanks for the suggestion. We have run clustering and dimensional reduction separately for real tumor and PDX, which was used for cell annotations (Figure S1A). We extracted tumor cells from real tumor and PDX and integrated them using Seurat (Figure 5A). By investigating the distribution of real tumor and PDX on the UMAP and bar plot (Figure 5B and S3C), we can see their overlapping and disparity for specific tumor subpopulations. For example, PDX has higher proportion of initiating/driving tumor cells (Tr-2), while real tumor has more progressive tumor cells (Tr-5). We agree that clustering separately could reveal more local features of tumor subpopulations, but we also think that cell clustering in the integrated data could better utilize the comprehensive knowledge of cells from multiple groups and capture the global features of tumor sub-populations. To allow visualization and greater comparison of the samples we have also included heatmaps with individual tumor, background and PDX samples in figure S6. Apart from the reduced dimension of the integrated data, we also interrogated gene signatures and enriched functions for each cluster of both sample groups (Figure 5C and 5F), which shows more holistic comparisons of real tumor and PDX. More details can be found in the result section.
R1 (2) In Figure 4, it is good to see the tumor cells and PDX cells showed high correlation. Yet I think it is still worthwhile to look at the genes that are not correlated, and discuss potential reasons for the differences, that will be helpful for anyone using PDX model when they interpret their data.	Thanks for the constructive suggestion. We followed your advice and conducted differential expression analysis and gene enrichment analysis for tumor cells in real tumor and PDX (Methods). Although the R square calculated for genes between real tumor and PDX (0.902) is pretty high (Figure 4A), we still observed distinct features between these two groups (Figure 4B, Table S3). We found that cell cycle genes are upregulated in PDX, which might be related with the changes made within tumor to support growth in this model. In contrast, tumor cells in real tumor displayed

	upregulated metabolic process and responses to external stimulus, which might be related with the interactions with tumor microenvironment which is lost in our PDX model. Heatmaps in Figure S6 also allow visualization of similarities and differences across samples.
R1 Technical points: (1) It will be good to provide more technical details, especially for the part covering cutting-edge scRNA sequencing data analysis. For example, the author mentioned certain cells are removed based on number of UMIs or MT reads, but without a discussion or reasoning what may occur to these cells that justify their removal. For another example, the authors said that the total UMI counts per cell were normalized to one million, without providing a raw UMI counts distribution in the cell population for the readers to visualize the data quality. Also by 10X Genomic platform it is possible to rely on RPKM as well, and there must be some reason to discuss why the authors chose to use UMI counts instead.	We have updated methodologies and added more detail and clarity to our approach, including human subjects, processing, tumor cell clustering, integration, differential analysis, and pathway enrichment. All additional details are provided in the methods section. We apologize that we didn't explain clearly about removal of some cells. In our analysis of new data, we applied more stringent quality control (cells with fewer than 500 expressed genes or 800 UMIs, or greater than 10% mitochondrial counts were removed) and we didn't find such a T3 group. We assume they were removed due to the stricter mitochondrial percentage threshold. Thank you for letting us know your concern about count normalization. In 10x Genomics, each transcript is tagged with a sequence serving as Unique Molecular Identifier (UMI). The gene-length bias does not exist. As a result, it's not recommended to normalize UMI count by gene length. More details can be found in their website: https://kb.10xgenomics.com/hc/en-us/articles/115003684783-Should-I-calculate-TPM-RPKM-or-FPKM-instead-of-counts-for-10x-Genomics-data Hence, we used the typical way of normalization in Seurat in our analysis https://satijalab.org/seurat/articles/pbmc3k_tutorial.html).
R1 (2) When doing clustering, the authors claimed that 1 cluster has been removed among the 10 identified clusters, due to some abnormality. I wonder if it is possible to remove such abnormal cells first, and perform clustering later, to see if indeed the remaining 9 clusters will still exist. Sometimes the existence of abnormal cells during clustering analysis may affect the outcome of other clusters.	Thanks for the careful observation. We totally agree with your point. With inclusion of more samples, we carefully investigated the quality of data and applied a more stringent QC (Methods) before the analysis was redone. Doublets and abnormalities were removed first and then clustering was applied (Methods). In the new analysis, we didn't find such a T3 group. We assume they were removed due to the stricter mitochondrial percentage threshold.

R1 (3) Please provide more detailed info about how the authors performed clustering analysis, for the readers to make a well-informed judgement. Is it PCA or t-SNE? Is the outcome robust when the parameters or the sample size fluctuate?	Thanks for bringing this up. It's a very good point and we have added more details in our Methods section based on your advice. We applied clustering for all cells (Figure 3A) and tumor cells (Figure 5A). Their strategies were pretty similar with minor difference of details. Here we copied a paragraph of clustering for all cells from the method section: Top 2,000 highly variable genes were selected using the "vst" method of FindVariableFeatures function in Seurat. Data was scaled and principal component analysis (PCA) was conducted using highly variable genes with functions ScaleData and RunPCA respectively. Principal components were used for integration of data from different samples using RunHarmony function of package Harmony. Neighbors were found with top 30 components of Harmony-corrected cell embeddings using shared nearest-neighbor (SNN) graph implemented in the function FindNeighbors. Clustering was conducted using Louvain algorithm. Different resolutions, including 0.5, 1 and 2, were used to identify both general and fine-grained clusters. Uniform Manifold Approximation and Projection (UMAP) was calculated in the PCA space for visualization in the reduced dimensions. More details can be found in the methods.
R2 (1) The analysis of clonal analysis based on low-resolution CNV analysis (based on tools such as inferCNV) is lacking and the results might be very helpful in tracking the subpopulations from a perspective that well complements the GE-based clusters.	Thank you for this suggestion. Based on the identification of clusters within combined tumor samples we determined the utilization of InferCNV analysis did not show a clear presentation of the chromosomal copy number variations including deletions and gains within the single nuclei isolated for our study. To best supplement this study the analysis would need to be performed on each cluster and perhaps separated by patient which we feel is outside the scope of this report. We appreciate the suggestion and for future investigations we will evaluate the utility of this methodology to further compare different tumors.
R2 (2) Although the functional roles of identified tumor clusters were discussed, its translational implications are not studied. There are many bioinformatics tools can be used to deconvolute bulk tumor based on the gene expression signatures derived from scRNAseq. For example, do patients with more T5/6 signatures in the bulk tumor has worse outcome?	This is a valid and important point, that we hope to explore with more samples than we have available for the current manuscript. Our future directions include exploration of the translational utility of scRNA sequencing of patient tumor and targeted treatment strategies tested in PDX models, but this work is still in progress and we anticipate need of additional samples.

Major comments: R3 1) First of all, the number of cases is small (n=1) which makes it very hard to judge whether the pattern presented is unique to this special case or it is rather general. It will be important to increase the number of cases.	We appreciate the feedback and recognize an increased N adds strength. More samples have been included to allow for a more robust analysis though a large N is challenging given the incidence of HB. We have added 2 more background samples (1 is discarded due to low sequencing quality), 2 more tumor samples (HB30 and HB53) and 1 more PDX tumor. More details can be found in Table S1, S2, Figure 1 and Methods.
R3 2) The authors presented a lot of results, many of which should have been explored by previous studies (e.g. bulk RNA analysis). There is not enough of link with previous work from the field and the novelty of the work is not high-lightened enough.	Thank you for pointing this out to us. Additional reference to prior HB rnaseq was included and utilized to guide exploration with single cell RNA seq in our samples. References include Sumazin, et. al., Hirsch, et. al., and Stafman, et.al. The key finding was stated with greater emphasis.
Detail comments: R3 3) The introduction of the manuscript is rather loose. The gap in the field and the scientific question is not very focused.	Apologies for not being precise in our writing. We have restructured to streamline the main focus and defined scientific question in the introduction and abstract. Extraneous details were removed from the introduction to clarify the key message.
R3 4) The experimental setup, especially the patient cohort and how they were used in different genomic survey (i.e. single cell or other approaches) is not easy to follow.	We have clarified utilization of patient samples for each methodology. Figure 1 diagram has been updated to clarify workflow and utilization of samples for single cell analysis to better display our experimental process.
R3 5) The authors tended to cherry pick different genes and pathways to interpret their findings. (e.g. GPC3, DLK1 and IGF2 on page 9, GPC3, YAP1 etc pathways on page 11). The logic behind the arguments is not so clear and it often made the presentation very punctuated.	We have clarified that pathways and markers are defined by expression analysis and the most differentially expressed pathways and genes were highlighted and from the DEGs we identified published genes of interested present in defined cell clusters. Our findings of pathways (WNT, C-Myc) and genes (GPC3, YAP1...) were based on differential expression analysis and gene enrichment analysis. More details can be seen in Figure 4, S3,4,5, and Table S3.
R3 6) The similarity between PDX and primary tumor is somewhat a controversial topic. There always will be some changes and mixed conclusions have been drawn for the differences. It will be good to draw the conclusion properly.	We understand and appreciate this point. We have highlighted histologic and gene expression similarities to validate prior published studies utilizing the heterotopic PDX model. We have revised conclusions to clearly state that the differences may be due to a number of reasons and further research is needed to validate these conclusions.

R3 7) The most novel observation is Figure 6. However, the interpretation is very abstract. The pathways displayed in Figure 6D and Figure 7 are not explained clearly. How the evolutionary trajectory depicted in Figure 6B was derived is not explained properly. With a bigger sample size, I think the study has the potential to draw a nice landscape of this important tumor.

Thank you for this observation. The most differentially expressed genes were used to map pathways in distinct tumor clusters and those clusters were subjected to network analysis from toppgene enrichment to demonstrate how key genes within tumor or specific clusters can be mapped to identify novel targets or treatment strategies following extensive validation. These genes and networks were used to show RNA velocity and predict the flow of gene expression within tumor.

Figure 1 (revised2)

Patient #	Sample Source	PRETEXT Stage	Histology	Invasion on Path	Metastasis	Patient Outcomes	Follow-Up (Days)	PDX Model
17	Liver	N/A	HB – epithelial (fetal, embryonal), macrotrabecular, poorly differentiated small cell with anaplastic features	Negative margins, vascular invasion	None	Deceased	139	X
18	Liver	N/A	HB - epithelial (fetal & embryonal), mesenchymal without teratoid features, HCC-like; 50% viable	Negative margins, lymphovascular invasion	None	Alive without disease	970	X
20	Liver	IV	HCC – Grade 3-4	Positive margin, vascular invasion, negative LN (0/1)	None	Deceased	171	
21	Liver	IV - P, E, F	HB - epithelial (embryonal), pleomorphic, HCC like; 80% viable	Focal positive margins, lymphovascular invasion, negative LN (0/3)	None	Alive without disease	946	X
22	Liver	III – F, C	HB - small foci of treated disease, no subtypes specified	Focal positive margin, LN negative (0/1)	None	Alive without disease	773	
23	Liver	III - P,V, C	HB – Mild pleomorphism, otherwise treated disease; 25% viable	Negative margins	None	Alive without disease	835	X
24	Liver	IV - V, P, E, F, C, M	HB – HCC-like, small cell undifferentiated, mesenchymal osteoid; 5-15% viable	Negative margins, focal tumor rupture, vascular invasion, LN negative (0/1)	Bilateral lungs	Alive with lung recurrence	742	
25	Liver	IV - V, P, F, M	HB – epithelial (fetal), osteoid; <10% viable	Focally positive margins with non-viable tumor, vascular invasion, LN negative (0/3)	Bilateral lungs	Alive without disease	805	
26	Llver	IV - F, C	HB – epithelial (embryonal, fetal), blastemal component; 15% viable	Negative margins, lymphovascular invasion, LN negative (0/1)	None	Alive without disease	790	
27	Liver	III- C, M	HB – epithelial (fetal, embryonal), osteoid; 45% viable	Negative margins, lymphovascular invasion, LN negative (0/1)	Bilateral lungs	Alive without disease	742	
28	Liver	III - M	HB – osteoid, epithelial (fetal), undifferentiated small cell; 15% viable	Margin positive, lymphovascular invasion, LN negative (0/2)	Bilateral lungs	Alive without disease	762	
30	Liver	IV - M	HB - epithelial (fetal, embryonal), transitional, small cell undifferentiated, HCC like, pleomorphic; 80% viable	Negative margins, lymphovascular invasion	Bilateral lungs	Alive with liver recurrence	670	X
31	Liver	II - M	HB - epithelial (fetal, embryonal), mesenchymal, with blastema ; 50% viable	Margin positive, lymphovascular invasion, LN negative (0/2)	Bilateral lungs	Alive with metastatic disease	664	
38	Liver	IV - P, V, F, C	HB - epithelial, mesenchymal with teratoid features. 50% viable	Negative margins, vascular invasion, LN negative (0/6)	None	Alive without disease	557	
53	Liver	IV - P, V	HB - epithelial (embryonal, pleomorphic) and HCC-like, focal mesenchymal with osteoid. 50% viable	Negative margins, vascular invasion, LN negative (0/5)	Bilateral Lungs	Alive without disease	269	

Figure 5 (revised-2)

Figure 6 (revised)

Figure 7 (Revised-2)

REVIEWERS' COMMENTS:

Reviewer #1 (Remarks to the Author):

The authors have addressed my concerns and suggestions, with the revised manuscript much improved in terms of data analysis and interpretation.

Reviewer #3 (Remarks to the Author):

The authors have addressed my questions.